# Therapeutic Efficacy of Mesenchymal Stem/Stromal Cell Small Extracellular Vesicles in Alleviating Arthritic Progression by Restoring Macrophage Balance

**DOI:** 10.3390/biom13101501

**Published:** 2023-10-10

**Authors:** Bin Zhang, Ruenn Chai Lai, Wei Kian Sim, Sai Kiang Lim

**Affiliations:** 1Institute of Molecular and Cell Biology (IMCB), Agency for Science, Technology and Research (A*STAR), 61 Biopolis Drive, Proteos, Singapore 138673, Singapore; zhang_bin@imcb.a-star.edu.sg (B.Z.); lai_ruenn_chai@imcb.a-star.edu.sg (R.C.L.); eugene_sim@imcb.a-star.edu.sg (W.K.S.); 2Paracrine Therapeutics Pte. Ltd., 10 Choa Chu Kang Grove #13-22 Sol Acres, Singapore 688207, Singapore; 3Department of Surgery, YLL School of Medicine, National University Singapore (NUS), 5 Lower Kent Ridge Road, Singapore 119074, Singapore

**Keywords:** mesenchymal stem/stromal cell (MSC), small extracellular vesicles (sEVs), rheumatoid arthritis (RA), collagen-induced arthritis (CIA), immunomodulation, M2 macrophage

## Abstract

Rheumatoid arthritis (RA) is a chronic autoimmune disease characterized by joint inflammation and damage, often associated with an imbalance in M1/M2 macrophages. Elevated levels of anti-inflammatory M2 macrophages have been linked to a therapeutic response in RA. We have previously demonstrated that mesenchymal stem/stromal cell small extracellular vesicles (MSC-sEVs) promote M2 polarization and hypothesized that MSC-sEVs could alleviate RA severity with a concomitant increase in M2 polarization. Here, we treated a mouse model of collagen-induced arthritis (CIA) with MSC-sEVs. Relative to vehicle-treated CIA mice, both low (1 μg) and high (10 μg) doses of MSC-sEVs were similarly efficacious but not as efficacious as Prednisolone, the positive control. MSC-sEV treatment resulted in statistically significant reductions in disease progression rate and disease severity as measured by arthritic index (AI), anti-CII antibodies, IL-6, and C5b-9 plasma levels. There were no statistically significant differences in the treatment outcome between low (1 μg) and high (10 μg) doses of MSC-sEVs. Furthermore, immunohistochemical analysis revealed that concomitant with the therapeutic efficacy, MSC-sEV treatment increased anti-inflammatory M2 macrophages and decreased pro-inflammatory M1 macrophages in the synovium. Consistent with increased M2 macrophages, histopathological examination also revealed reduced inflammation, pannus formation, cartilage damage, bone resorption, and periosteal new bone formation in the MSC-sEV-treated group compared to the vehicle group. These findings suggest that MSC-sEVs are potential biologic disease-modifying antirheumatic drugs (DMARDs) that can help slow or halt RA joint damage and preserve joint function.

## 1. Introduction

Rheumatoid arthritis (RA) is a chronic autoimmune disorder marked by persistent synovial inflammation, accompanied by cartilage degradation and bone erosion. These processes culminate in the development of joint deformities and a decline in functional capacities [1,2]. While current treatment strategies to mitigate inflammation and manage symptoms are effective for many patients, many also do not respond well [3,4]. Notably, extended use of a widely used corticosteroid in RA treatment, prednisolone, has been associated with heightened mortality rates [5,6]. Therefore, there is a critical need to develop innovative and safe therapeutic approaches that can effectively target the fundamental mechanisms driving the progression of RA.

In this regard, mesenchymal stromal cells (MSCs) are a highly promising therapeutic tool to address RA, primarily due to their ability to modulate the immune response [7,8]. Although MSCs were initially thought to home injured tissues, engraft, and differentiate into appropriate cell types for tissue repair, they are now thought to exert their therapeutic activity primarily through secreted factors rather than direct differentiation [9,10].

In the investigation of MSC secretion, we were first to report that the therapeutic agent in MSC secretion had a molecular weight larger than 1000 kD [11]. Subsequently, Cammussi and his colleagues referred to this therapeutic agent as “microvesicles”, with a size range of 80–1000 nm [12], while we identified it as “exosomes” with a size range of 100–130 nm [13]. Both microvesicles and exosomes are extracellular vesicles (EVs). In 2017, Cammussi and colleagues further revealed that the smaller ~160 nm fraction, rather than the larger ~215 nm fraction, exhibited the most significant therapeutic potential [14]. Consequently, it is now widely acknowledged that the therapeutic activity of MSCs is primarily mediated by small EVs (sEVs) in the 50–200 nm size range [15]. Importantly, MSC-sEVs have shown therapeutic efficacy comparable to their parental MSCs [12,16,17,18].

MSC-sEVs can be prepared using different sources of MSCs such as bone marrow or adipose and EV enrichment protocols such as ultracentrifugation and size exclusion chromatography. Although most shared a core set of critical quality attributes [19], their proteomes can vary according to differences in MSC sources and preparation methods [20]. To mitigate such variations, we used an immortalized clonal MSC to prepare our MSC-sEV preparations [21]. These MSC-sEV preparations have a diverse proteome with over 200 immunomodulatory proteins and they exhibit a spectrum of immunomodulatory activities such as inhibition of mitogen-activated lymphocyte proliferation, secretion of anti-inflammatory cytokines, promotion of regulatory T cell (Treg) and M2 macrophage polarization, inhibition of C5b9 complement complex formation, and suppression of C5b9-induced NETs and neutrophil secretion of IL-17 [22,23,24,25,26,27,28,29,30,31].

Given the correlation between an enhanced presence of anti-inflammatory M2 macrophages and therapeutic response in RA [32], we postulate that our MSC-sEV preparations with their established efficacy in promoting M2 macrophage polarization [23,31,33] could alleviate the severity of RA. To test this hypothesis, we employed a mouse model of collagen-induced arthritis (CIA), a widely used preclinical model that mimics essential pathological features of human RA [34,35]. Our evaluation involved measuring the Arthritic Index (AI) scores and systemic inflammation levels, examining histopathological changes, and assessing macrophage polarization in both control and treated animals.

## 2. Materials and Methods

### 2.1. Culture of MSC and Preparation of MSC-sEV

Immortalized E1-MYC 16.3 human ESC-derived mesenchymal stem cells were cultured in DMEM with 10% fetal calf serum, as previously described [21]. For MSC-sEV preparation, the cells were grown in a chemically defined medium for three days until 80% confluence was reached [13]. The defined medium was made by mixing 480 mL DMEM (31053, Thermo Fisher, Waltham, MA, USA) with 5 mL each of NEAA (11140-050, Thermo Fisher, Waltham, MA, USA), L Glutamine (25030-081, Thermo Fisher, Waltham, MA, USA), Sodium Pyruvate (11360, Thermo Fisher, Waltham, MA, USA) and ITS-X (51500-056, Thermo Fisher, Waltham, MA, USA), as well as 0.5 mL of 2-ME (21985-02, Thermo Fisher, Waltham, MA, USA). Additionally, 0.1 mL of bFGF (0.5 ng/μL 0.2%BSA in PBS (+)) and 0.005 mL of PDGF (100 ng/μL PBS (+)) were added into the mixture, which were obtained separately as follows: bFGF (13256-029, Thermo Fisher, Waltham, MA, USA), Bovine Serum Albumin (BSA, A9647, Sigma-Aldrich, St. Louis, MO, USA), PDGF (100-00 AB CYTOLAB, Karlsruhe, Germany) and PBS(+) (14040-133, Thermo Fisher, Waltham, MA, USA). The conditioned medium (CM) was tangentially flow filtered and then concentrated 50 times using a membrane with a molecular weight cut-off (MWCO) of 100 kDa (Sartorius, Gottingen, Germany). The protein concentration of the MSC-sEV preparation was measured using a Coomassie Plus (Bradford) Assay Kit (ThermoFisher Scientific, Waltham, MA, USA). Only batches of exosomes determined by Nanoparticle tracking analysis on a ZetaView instrument (Particle Matrix GmbH, Dusseldorf, Germany) that met specific parameters, such as 1.46 × 10^11^ ± 2.43 × 10^10^ particles per ug protein and particle modal size of 138.62 ± 4.45 nm, were used for this study. Additionally, the sEV preparations were filtered with a 0.22 μm filter (Merck Millipore, Billerica, MA, USA) and determined by Western blot or ELISA to express CD81 and CD73 before being stored in a −80 °C freezer.

### 2.2. Mice

Fifty-five DBA/1J mice were purchased from Jackson Laboratories (male, 7–9 weeks old) for the CIA experiments, individually examined and housed in five cages of ten mice each, and one cage containing five mice. The mice were placed in quarantine with daily inspections, and were ear-tagged with unique numbers for identification purposes.

### 2.3. CIA Model

This study was performed by Washington Biotechnology INC, 6200 Seaforth Street Baltimore, MD 21224 under IACUC no: 21-059.11. For making 0.01 M acetic acid, 0.1 mL glacial acetic acid (Cat. A35-500, Thermo Fisher Scientific, Waltham, MA, USA) was dissolved in 160 mL de-ionized water. After that, 20 mg bovine type II collagen (BCII, Cat. 20021, Chondrex, Inc., Woodinville, WA, USA) was solubilized to a concentration of 4 mg/mL in 5 mL 0.01 M acetic acid at 4 °C with constant mixing overnight. On Day 0, the 3 mL BCII was emulsified with 3 mL Complete Freund’s adjuvant (CFA, Cat.7001, lot 200224, Chondrex, Inc., Woodinville, WA, USA, 4 mg/mL *Mycobacterium tuberculosis*) on an ice bath in six batches, the mice were weighed, and fifty mice were immunized subcutaneously (SC) at the base of the tail with 50 µL of the BCII/CFA emulsion. Five mice were not inoculated and were identified as Naïve control. On Day 21, the 3 mL BCII was again emulsified with 3 mL Incomplete Complete Freund’s adjuvant (ICFA, Cat. F5506, lot SLBT9741, Sigma-Aldrich, St. Louis, MO, USA) on an ice bath in six batches, and the mice were weighed and boosted SC at the base of the tail with 50 µL of the BCII/ICFA emulsion. Between DAY 22–52, the mice were weighed and scored for signs of arthritis daily as defined in Table 1. Each paw was scored, and the sum of all four scores was recorded as the Arthritic Index (AI). The maximum possible AI was 16. On Day 28, the treatment regimen was initiated. Every ten mice upon scoring an average AI of 2.8 were assigned to a treatment group. Treatment was initiated on the day of assignment as per Table 2. After 14 days of therapy, the mice in each group were weighed and scored for signs of arthritis. The mice in each group were anesthetized and exsanguinated into pre-chilled EDTA-treated tubes (Cat. 365974, Becton Dickinson, Franklin Lakes, NJ, USA), respectively. The blood samples were processed to plasma which was stored in six aliquots in labeled Eppendorf tubes at −80 °C. The limbs were individually removed to 10 mL neutral buffered 10% formalin (Cat. 5701, Richard-Allan Scientific, San Diego, CA, USA) and the carcasses were disposed of appropriately.

### 2.4. Mouse Plasma Anti-CII Antibody and Cytokine Measurement

The plasma aliquots were thawed to room temperature. The samples for Naïve control and the other groups were diluted 1:1000 and 1:50,000, respectively, and then assayed by ELISA for mouse Type II collagen-specific IgG (anti-CII) antibody (Cat. 2036T, Chondrex, Inc., Woodinville, WA, USA). The remaining samples were diluted 1:2 and assayed by ELISA for IL-6 (Cat. M6000B, R&D Systems, Minneapolis, MN, USA), IL-10 (Cat. M1000B, R&D Systems, Minneapolis, MN, USA), IL-17 (Cat. M1700, R&D Systems, Minneapolis, MN, USA), IL-23 (Cat. M2300, R&D Systems, Minneapolis, MN, USA), and C5b-9 (Cat. OKCV01374, Aviva Systems Biology, San Diego, CA, USA). The results were analyzed using Student’s *t*-test. *p* values < 0.05 were considered as statistically significant.

### 2.5. Histological Assessment of CIA

The formalin-preserved limbs were performed for histopathological processing and evaluation. Paws were embedded in paraffin in the frontal plane. Ankles, if left attached to the hind paw, were also embedded in the frontal plane but may have been detached and sectioned in the sagittal plane for special purposes. Sections were cut and stained with toluidine blue. The methods for toluidine bluejoint staining are a modified version of those described by Schmitz [36]. For the evaluation of joints, when scoring paws or ankles from mice with lesions of type II collagen arthritis, severity of changes as well as number of individual joints affected must be considered. When only one to four-digit joints and/or the wrist/carpus or ankle/tarsus are affected, an arbitrary assignment of a maximum score of 0.5, 1, 2, or 3 for the parameters below is given depending on severity of changes as described in Table 3. If more than four-digit joints plus wrist/ankle are involved, the assigned scores are generally 4 or 5 for inflammation and will vary according to the criteria (Table 3) for the other parameters. The inflammatory infiltration in mice with type II collagen arthritis consists of neutrophils and macrophages with smaller numbers of lymphocytes when the lesions are in the acute to subacute phase. Tissue edema and neutrophil exudates within the joint space are common in the acute to subacute phase. As the inflammation progresses to chronic, mononuclear inflammatory cells (monocytes, lymphocytes) predominate and fibroblast proliferation, often with deposition of metachromatic matrix, occurs in synovium and periarticular tissue. Exudate is less common in the joint space. Unless indicated in the comments area, the inflammation type is acute to subacute. DBA mice have an increased incidence of dactylitis and onchyoperiostitis affecting the nail bed and distal phalynx [37]. These lesions were recorded in the comment section but were not included in the inflammation score. The associated parameters were scored according to the indicated criteria in Table 3.

### 2.6. Immunohistochemistry Staining for CD163

As a marker of M2 macrophages, CD163 was assessed [38] by standard immunohistochemistry (IHC), and CD86 was also assessed as a marker for M1 macrophages [39]. For IHC staining, Immunocal (StatLab #1414) decalcified formalin-fixed, paraffin-embedded (FFPE) mouse paws, staining was conducted on a Leica Bond RX platform using standard chromogenic methods. The sections were cut to 5 μm thickness. For antigen retrieval, slides were incubated with a pH6 Tris-based buffer (CD86) or a pH9 EDTA-based buffer (CD163) for 2 h at 70 °C. Subsequently, a 5-min enzyme block with 3% H2O2 and a 10-min protein block with 1% Casein were conducted at room temperature, followed by a 30-min antibody incubation (CD86—1:400, Cell Signaling rabbit clone E5W6H, # 19589; CD163—1:400, Abcam rabbit clone EPR19518, ab182422) at room temperature. Antibody binding was detected using an HRP-conjugated, anti-rabbit secondary polymer, followed by chromogenic visualization with diaminobenzidine (DAB). A Hematoxylin counterstain was used to visualize nuclei. For quantification analysis of CD163 and CD86, positively stained cells were counted, respectively, in five randomly selected fields in the most severely inflamed joints using a micrometer to delineate an area that was 10 × 100 units at 400× magnification. Two ankle joints at most were counted, then digit joints were counted. Comments were also made about the location of immune-positive cells within the exudate/infiltrate in the joint space and expanded synovial lining/subsynovial tissue. The subsynovial areas of the inflamed synovium, excluding the lining or exudate, exhibited the highest numbers of CD163-positive macrophages. In contrast, normal or near-normal joints showed higher numbers of CD163-positive macrophages in the synovial lining. Therefore, counts were conducted in both areas to highlight the distribution differences between normal and diseased joints.

### 2.7. Statistical Tests

Statistical analyses were performed using Bonferroni Multiple Comparison Test on GraphPad Prism 6 (GraphPad Software Inc., La Jolla, CA, USA). *p* values < 0.05 were considered statistically significant. The average value and standard deviation of each group were calculated by the individual animal in the group. A trend or tendency was assumed when a one-tailed *t*-test returned *p* values < 0.1. Results were expressed as mean ± SD.

## 3. Results

### 3.1. MSC-sEVs Alleviate the Arthritic Index (AI) in a Mouse Model of CIA

The first signs of disease were observed six days after the second immunization with bovine type II collagen (BCII) (Figure 1). As animals developed the disease, they were sorted into treatment groups with an average AI of 2.8 (range of 2–4) prior to the initiation of the dosing regimen (Table 1). For MSC-sEV treatment, each CIA mouse was injected intraperitoneally (IP) with 1 or 10 μg MSC-sEVs, as described in the Materials and Methods (Figure 1, Table 2). Prednisolone, a corticosteroid that is used to treat RA, was administered orally as a positive control [40] (Figure 1, Table 2).

Compared to the naïve (Normal) mice, the induction of CIA caused a significant reduction in body weight (Figure 2). However, there were no significant differences in mean body weight among the Vehicle group, Prednisolone group, and the two MSC-sEV groups during the treatment period (Figure 2).

We assessed disease progression by AI as defined in (Table 1). All CIA animals showed elevated AI with statistically significant differences among the groups (Figure 3). The Vehicle group had an AI of 12.2 after fourteen days. Prednisolone treatment arrested disease progression and reduced disease severity with a significant 95% reduction at fourteen days. MSC-sEVs were less effective than prednisolone in reducing the rate of disease progression and yielded a significant 42–50% reduction in disease severity at fourteen days (Figure 3). The AI was not statistically significant different between 1 and 10 μg doses of MSC-sEVs, indicating that the maximum therapeutic dose was 1 μg MSC-sEVs per mouse.

### 3.2. MSC-sEVs Decrease Anti-CII, IL-6, and C5b-9 in a Mouse Model of CIA

Additionally, we analyzed the plasma for antibody to mouse Type II collagen (anti-CII), C5b-9, IL-6, IL-17, IL-10, and IL-23 using commercially available ELISA kits. We found that the CIA mice exhibited a significant increase in terminal plasma concentrations of anti-CII, C5b-9, IL-6, and IL-17, relative to the normal mice (Figure 4). In the Prednisolone group, the terminal plasma concentration of IL-6 was significantly reduced by 77% (Figure 4). For the MSC-sEV treated mice, the 1 µg/mouse dose yielded a significant inhibition of terminal plasma IL-6, while the 10 µg/mouse dose yielded a significant inhibition of terminal plasma concentration of anti-CII, IL-6 and C5b-9 (Figure 4). Despite these differences, the average AI scores in both groups of mice were not statistically different.

### 3.3. MSC-sEVs Reduce the Histopathological Scores in a Mouse Model of CIA

To investigate if the improved AI score in MSC-sEV treated mice (Figure 3) was underpinned by improved histopathological changes, the paws of the 1 µg µg/mouse treatment group (n = 10) analyzed both normal (n = 1) and CIA mice (n = 10) (Figure 5A) and scored for the presence of inflammation, pannus, cartilage damage, bone resorption, and periosteal new bone formation, as defined in Table 3. The MSC-sEV group had statistically significant lower scores than the Vehicle group, namely, reduced paw inflammation (56% reduction), pannus formation (77%), cartilage damage (67%), bone resorption (77%), periosteal bone formation (63%), and summed paw scores (67%) (Figure 5B).

### 3.4. MSC-sEVs Increase Anti-Inflammatory M2 but Not Pro-Inflammatory M1 Macrophages

We had previously reported that MSC-sEVs preferentially polarized M2 over M1 macrophages [23] through a CD73-mediated AKT phosphorylation pathway [33]. To determine if the efficacy of MSC-sEVs in alleviating the severity of CIA was concomitant with increased M2 macrophage polarization, we assessed the relative abundance of CD163^+^ M2 and CD86^+^ M1 macrophages by immunohistochemistry (IHC) staining in paw sections.

In naïve mice, there was no detectable level of CD86^+^ or CD163^+^ cells in all layers of the synovium except synovial lining where there was abundant CD163^+^ cells. In the vehicle group, CD86^+^ cells were detected in all layers of the synovium, while CD163^+^ cells were present in the subsynovial tissue but not the synovial lining (Figure 6).

In contrast, both CD86^+^ and CD163^+^ staining was detected in all layers of the synovium in MSC-sEV treated mice and they exhibited a trend towards the staining pattern of the naïve mice than the vehicle controls. Relative to the vehicle animals, there was a 90% reduction of CD86^+^ in the synovium and 70% reduction of CD163^+^ cells in the subsynovial tissue (Figure 6). Concomitantly, CD163^+^ cells which were not detected in the synovial lining of vehicle-treated animals were elevated towards the level observed in naïve animals. Together, our data indicated that the relative distribution of CD86^+^ and CD163^+^ cells in MSC-sEV treated animals was trending towards the normal baseline distribution observed in naïve animals.

## 4. Discussion

In this study, we evaluated the therapeutic potential of MSC-sEVs in a mouse model of collagen-induced arthritis (CIA), a widely used preclinical model that mimics key pathological features of human RA [34,35]. Although current treatment strategies of suppressing inflammation and managing symptoms with steroids such as prednisolone are effective, the long-term use of prednisolone is associated with increased mortality [5]. Consistent with its effectiveness, daily oral administration of prednisolone resulted in an immediate arrest of disease progression and a significant reduction in disease severity. IP injection of MSC-sEVs also alleviated disease progression, terminal disease severity, systemic inflammation, and other immune responses associated with arthritis. Generally, MSC-sEV treatment was efficacious in reducing the severity of CIA but was not as effective as prednisolone. Nevertheless, MSC-sEVs could significantly mitigate joint damage and promote tissue repair in the arthritic joints.

Similar to MSCs, which are known to promote M2 macrophage polarization [41,42,43], MSC-sEVs also exhibit the same ability [23,44]. It is crucial for MSC-sEVs to alleviate disease severity in different animal studies, such as hyperoxia-induced lung injury [45], as well as bone and cartilage repair [29,30]. Consistent with our hypothesis that MSC-sEVs could mitigate the severity of RA by enhancing M2 macrophage polarization, we observed that the reduced disease severity in MSC-sEV treatment was concomitant with reduced pro-inflammatory CD86^+^ cells M1 macrophages in the synovium, and increased CD163^+^ anti-inflammatory M2 macrophages in the synovial lining.

MSC-sEVs have potential for treating RA due to their unique characteristics and mechanisms of action. They can traverse biological barriers and deliver their cargo directly to inflamed joints, enabling targeted therapy [46]. As RA is a highly complex autoimmune disease, the therapeutic effects of MSC-sEVs in RA are likely to be multifactorial and involve many aspects of the immune system. In addition to macrophages, RA is also associated with the dysregulation of several immune cell types such as Tregs [47,48] and neutrophils [49]. Humoral factors, such as complements, are also complicated in RA pathology [50]. To gain a comprehensive understanding of the immunomodulatory mechanisms underlying the function of MSC-sEVs in the context of CIA, it is essential to explore the intricate interplay between different immune compartments and the diverse immune modulating activities of MSC-sEVs. Previous investigations have revealed that MSC-sEVs can activate TLR4 and elicit an anti-inflammatory response in human and mouse primary monocytes [23]. Additionally, in vitro experiments have demonstrated that these sEVs can promote the polarization of naïve CD4^+^ T cells into CD4^+^CD25^+^Foxp3^+^ Tregs in the presence of allogeneic CD11c+ APCs [22]. In vivo studies have also shown that MSC-sEVs can augment Tregs in animals with allogenic skin grafts or GVHD [22,23], indicating their potential to induce regulatory cell types in immunologically challenged hosts [23]. Moreover, MSC-sEVs exhibit the capacity to modulate soluble humoral factors and innate immune cells by inhibiting the formation of the terminal complement complex through the presence of CD59 on MSC-sEVs [24]. This inhibition subsequently hinders complement-activated neutrophils, leading to a reduction in NETs and IL-17 both in vitro [27] and in vivo [26]. Additionally, the regenerative potential of MSC-sEVs to repair and restore joint integrity may also contribute to the efficacy of MSC-sEVs in treating RA [51].

In summary, our study’s findings underscore the therapeutic potential of MSC-sEVs in effectively mitigating RA, offering a promising avenue for treatment through immune modulation rather than mere immune suppression.

## Figures and Tables

**Figure 1 biomolecules-13-01501-f001:**
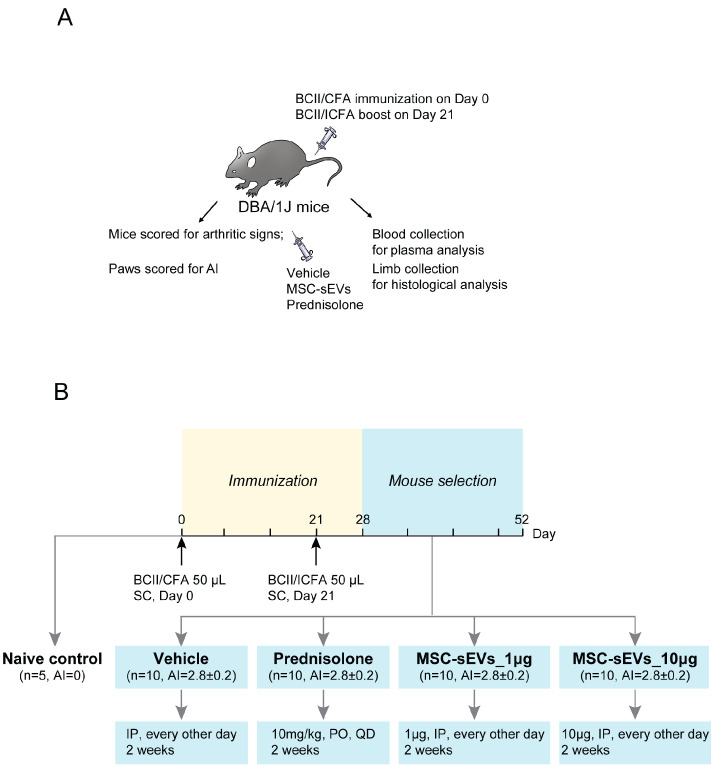
Summary of administration in a mouse model of collagen-induced arthritis (CIA). The diagram illustrates the experimental design (**A**) and treatment regimen (**B**). BCII was solubilized in 0.01 M acetic acid, and the BCII/CFA emulsion was prepared by emulsifying BCII with CFA on Day 0. Subcutaneous immunization with BCII/CFA emulsion was performed on fifty mice, while five mice were kept as Naïve controls. On Day 21, a boost was given using BCII/ICFA emulsion. The mice were monitored for signs of arthritis and weighed from Days 22 to 52. The AI was calculated based on the scores of each paw. On Day 28, the treatment regimen was initiated. The vehicle group received IP injections of the vehicle solution at 50 μL per mouse every other day, while the prednisolone group received daily oral administration of prednisolone at a dose of 10 mg/kg daily as a positive control. Both MSC-sEV groups were assigned and received IP injections of MSC-sEVs at doses of one or 10 μg/50 μL/mouse every other day, respectively. After 2 weeks of therapy, the mice were weighed, scored for signs of arthritis, and euthanized. Blood samples were collected for plasma analysis, and the limbs were fixed in neutral buffered 10% formalin for histological analysis. BCII: Bovine type II collagen; CFA: Complete Freund’s adjuvant; ICFA: Incomplete Complete Freund’s adjuvant; AI: Arthritic Index; SC: subcutaneous injection; PO: oral administration; IP: intraperitoneal administration; QD: once daily.

**Figure 2 biomolecules-13-01501-f002:**
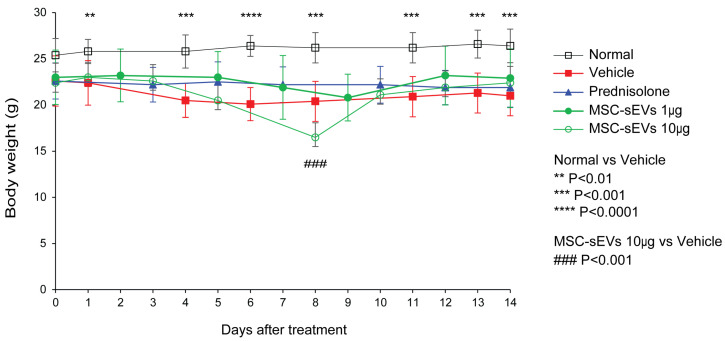
Evaluation of body weight in CIA mouse model. The body weight per mouse in each group (Naïve, Vehicle, Prednisolone, MSC-sEVs_1µg, MSC-sEVs_10 µg) was recorded daily after the treatment regimen and animals were sacrificed after 2 weeks of therapy. The body weight was analyzed using Student’s *t*-test. *p* values < 0.05 were considered statistically significant.

**Figure 3 biomolecules-13-01501-f003:**
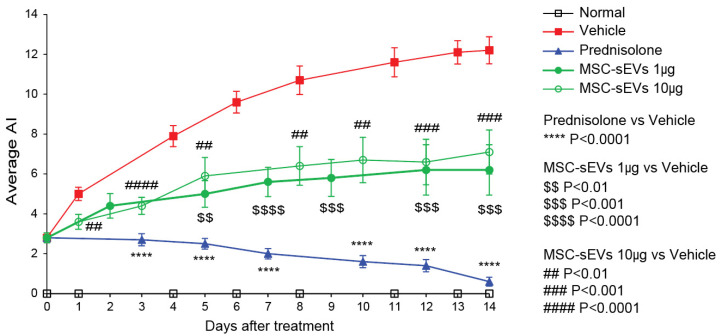
Evaluation of signs of arthritic progression. After subcutaneous immunization of mice with BCII/CFA emulsion and boost with BCII/ICFA emulsion, the mice were monitored for signs of arthritic progression. The AI was calculated and compared based on the scores of each paw of mice in the different groups, including Naïve control, Vehicle control, Prednisolone control, MSC-sEV_1µg, and MSC-sEV_10µg, during the period of 2 weeks of therapy. Each paw was scored on a scale from 0 to 4, and the sum of all four scores was recorded as the AI. The maximum possible AI was 16. The average AI was analyzed using Student’s *t*-test. *p* values < 0.05 were considered statistically significant.

**Figure 4 biomolecules-13-01501-f004:**
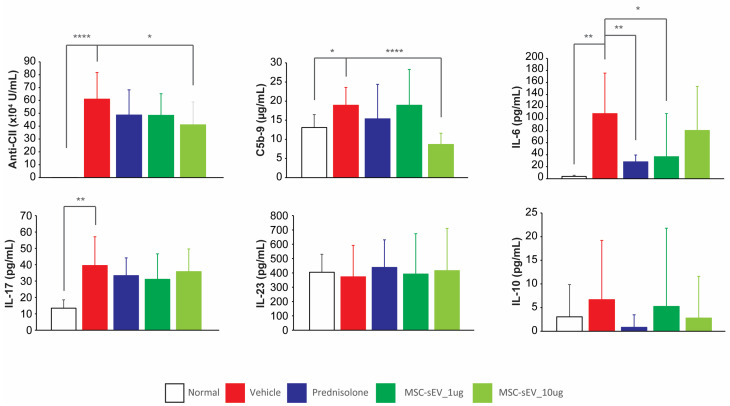
Measurement of plasma anti-CII, C5b-9, and associated cytokines in the CIA mouse model. The plasma aliquots were thawed to room temperature. For the Naïve control group, plasma samples were diluted 1:1000, while for the other treatment groups, samples were diluted 1:50,000. After that mouse anti-CII levels were measured by ELISA. The remaining samples were diluted 1:2 and assayed using the appropriate ELISA kits for C5b-9, IL-6, IL-17, IL-23, and IL-10. Statistical analysis was performed using Student’s *t*-test, and *p* values < 0.05 were considered as statistically significance. * *p* < 0.05, ** *p* < 0.01, **** *p* < 0.0001.

**Figure 5 biomolecules-13-01501-f005:**
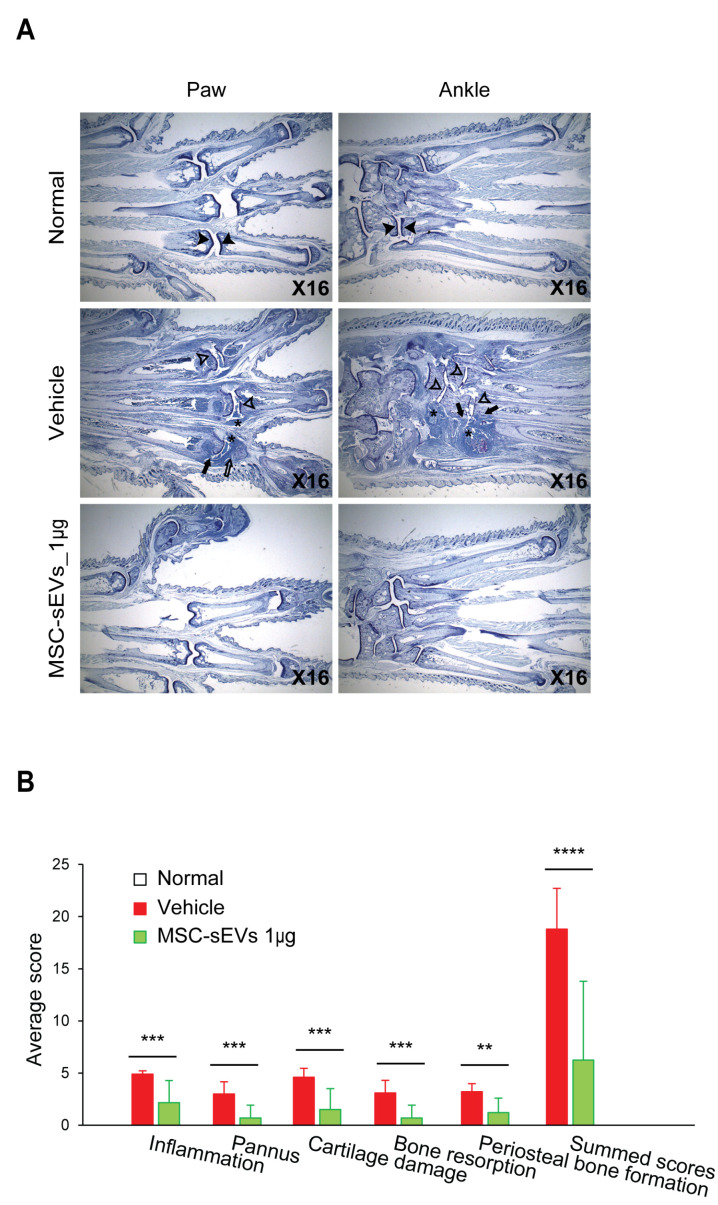
Evaluation of anti-arthritis mediated by MSC-sEVs in the CIA mouse model. (**A**) Representative photomicrographs of toluidine blue-stained joint sections are shown. The sections were cut from paraffin blocks of limb tissue prefixed in formalin solution and stained with toluidine blue. The symbol ‘*’ denotes representative areas of synovial inflammation. Closed arrows mark typical regions of pannus and bone resorption, while the open arrow indicates a region of complete cartilage and bone loss. Closed arrowheads indicate typical normal cartilage surfaces, whereas open arrowheads indicate thinned or damaged cartilage surfaces. (**B**) The score of arthritic lesions was calculated according to the criteria of Table 3. The bright field image of toluidine blue-stained sections was captured using a digital camera at 16-fold magnifications. For evaluation of joints (paws or ankles), severity of changes as well as number of individual joints affected were estimated. The associated parameters, such as inflammation, pannus, cartilage damage, bone resorption, and periosteal bone formation, were scored as per the indicated criteria in Table 3. A summary of the five histopathological scores is also calculated for each joint, and the cumulative score for these five parameters constituted the Summed Score. ** *p* < 0.01, *** *p* < 0.001, **** *p* < 0.0001.

**Figure 6 biomolecules-13-01501-f006:**
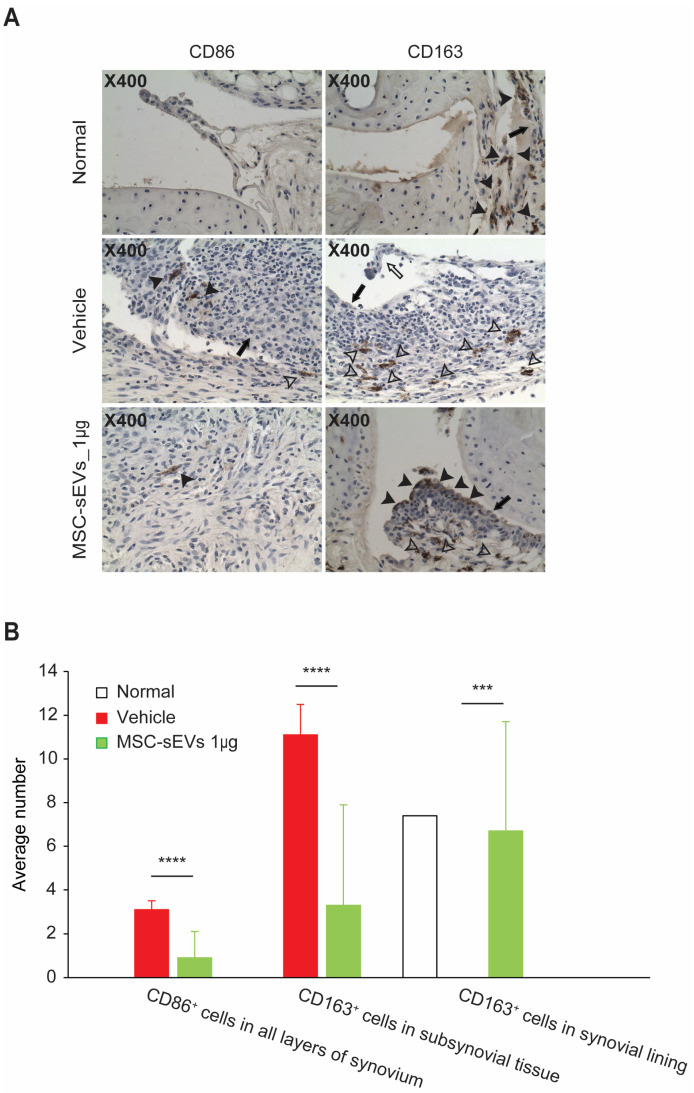
Assessment of M1 and M2 macrophage distribution in arthritic joints. IHC was performed to assess the presence of M1 and M2 macrophages in the arthritic joints. CD163, a marker for M2 macrophages, and CD86, a marker for M1 macrophages, were evaluated. Representative photomicrographs of CD163/CD86-immunohistochemistry-stained joint sections of CIA mice are shown ((**A**), upper panel). For quantification analysis, positively stained cells were counted in five randomly selected fields within the most severely inflamed joints and expressed as mean number of cells per high power field ((**B**), lower panel). A micrometer was used to delineate an area that measured 10 × 100 units at 400× magnification. The counting process included two ankle joints and then digit joints. Additionally, the location of immune-positive cells within the joint space and expanded synovial lining/subsynovial tissue was noted. The closed arrow identifies synoviocytes in the synovial lining. The open arrow identifies some synovial lining that separated during sectioning (artifact). Closed arrowheads identify immune-positive cells in synovial lining, and open arrowheads identify immune-positive cells in subsynovial tissue. Data represent mean ± SD. *p* values < 0.05 were considered as statistically significant. *** *p* < 0.001, **** *p* < 0.0001.

**Table 1 biomolecules-13-01501-t001:** Definition of arthritic score components.

Arthritic Extent	Score
No visible effects of arthritis	0
Edema and/or erythema of one digit	1
Edema and/or erythema of 2 joints	2
Edema and/or erythema of more than 2 joints	3
Severe arthritis of the entire paw and digits including limb deformation and ankylosis of the joint.	4

**Table 2 biomolecules-13-01501-t002:** Administrations in the CIA mouse model.

Group	No. Mice	AI	Dose	ROA	Regimen
Naïve	5	0	N/A	N/A	N/A
Vehicle	10	2.8 ± 0.2	50 µL/mouse	IP	Every other day for 2 weeks
Prednisolone *	10	2.8 ± 0.2	10 mg/kg	PO	Daily for 2 weeks
MSC-sEV_1 µg	10	2.8 ± 0.2	1 µg/mouse	IP	Every other day for 2 weeks
MSC-sEV_10 µg	10	2.8 ± 0.2	10 µg/mouse	IP	Every other day for 2 weeks

* The 53.16 mg prednisolone 21-hemisuccinate sodium salt (Cat. P4153, lot BCBB6186V, Sigma-Aldrich, St. Louis, MO, USA) was dissolved in 40 mL de-ionized water to prepare a 1 mg/mL solution. N/A: not applicable; PO: oral administration; IP: intraperitoneal administration.

**Table 3 biomolecules-13-01501-t003:** Histopathological evaluation of Joints in the CIA mouse model.

	Score	
**Inflammation**	0	Normal
	0.5	Very minimal, affects synovium of only one joint, either a digit joint or the wrist or ankle with minimal infiltration or there may be very minor multifocal synovial or periarticular infiltration of few inflammatory cells in a few joints
	1	Minimal, infiltration of inflammatory cells in synovium and periarticular tissue of 1–2 affected digit joints is generally minimal to moderate or minimal in wrist/ankle, generally about 1–10% of the area at risk is affected
	2	Mild, infiltration of inflammatory cells in synovium and periarticular tissue of 2–3 affected joints is generally mild to marked or mild in wrist/ankle, generally about 11–25% of the area at risk is affected
	3	Moderate, infiltration of inflammatory cells in synovium and periarticular tissue in up to 4 affected digit joints +/− wrist or ankle is generally mild to marked, generally about 26–50% of the area at risk is affected
	4	Marked infiltration of inflammatory cells in most joints with marked edema, several unaffected digit joints may be present, generally about 51–75% of the area at risk is affected
	5	Severe diffuse infiltration with severe edema affecting greater than 75% of all joints and periarticular tissues, greater than 75% of area at risk affected severely, a few digit joints may be less severely affected with not more than 1 unaffected digit joint
**Pannus**	0	Normal
	0.5	Very minimal, marginal zone only, less than 1% of area at risk affected
	1	Minimal infiltration of pannus in cartilage and subchondral bone, marginal zones mainly. Approximately 1–10% of area at risk affected
	2	Mild infiltration with marginal zone destruction of hard tissue in affected joints, 11–25% of area at risk affected
	3	Moderate infiltration with moderate hard tissue destruction in affected joints, 26–50% of area at risk affected
	4	Marked infiltration with marked destruction of joint architecture, affecting most joints, 51–75% of area at risk affected
	5	Severe infiltration associated with total or near total destruction of joint architecture, affects all joints, greater than 75% of area at risk affected
**Cartilage damage**	0	Normal
	0.5	Very minimal, affects marginal zones only of one to several joints, proteoglycan loss mainly
	1	Minimal, generally minimal damage with 1–10% cartilage loss in paws
	2	Mild, generally mild loss of toluidine blue staining (proteoglycan) with focal areas of chondrocyte loss and/or collagen disruption in some affected joints/areas with 11–25% overall cartilage loss in paws
	3	Moderate, generally moderate loss of toluidine blue staining (proteoglycan) with multifocal chondrocyte loss and/or collagen disruption in affected joints/area with 26–50% overall cartilage loss in paws
	4	Marked, marked loss of toluidine blue staining (proteoglycan) with multifocal marked (depth to deep zone or tidemark) chondrocyte loss and/or collagen disruption in most joints with a few unaffected or mildly affected with 51–75% overall loss in paws
	5	Severe, severe diffuse loss of toluidine blue staining (proteoglycan) with severe (depth to tide mark) chondrocyte loss and/or collagen disruption in most or all joints, greater than 75% loss in paws
**Bone Resorption**	0	Normal
	0.5	Very minimal resorption affects only marginal zones
	1	Minimal approximately 1–10% of area at risk of subchondral bone affected
	2	Mild, more numerous areas of resorption, approximately 11–25% of total area at risk of subchondral bone affected
	3	Moderate, obvious resorption of subchondral bone resulting in approximately 26–50% of area at risk of subchondral bone affected
	4	Marked, very obvious resorption of subchondral bone resulting in approximately 51–75% of area at risk of subchondral bone affected
	5	Severe, distortion of entire joint due to destruction approximately 76–100% of area at risk of subchondral bone affected
**Periosteal new bone formation**	0	Normal, no periosteal proliferation
	0.5	Minimal focal or multifocal early proliferation, measures less than 40 μm width (<1 unit on 25×)
	1	Minimal multifocal early proliferation, measures 40–80 μm width (1–2 units on 25×)
	2	Mild multifocal to diffuse with widths that measure approximately 120–200 μm (3–5 units on 25×)
	3	Moderate diffuse with widths that measure 240–280 μm (6–7 units on 25×)
	4	Marked diffuse with widths that measure 320–400 μm (8–10 units on 25×)
	5	Severe, diffuse with widths that measure greater than 400 μm (>10 units on 25×)

## Data Availability

The data presented in this study are all available in this article.

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
