# Peer review of "Therapeutic Efficacy of Mesenchymal Stem/Stromal Cell Small Extracellular Vesicles in Alleviating Arthritic Progression by Restoring Macrophage Balance"

_biomolecules, 2023, doi:10.3390/biom13101501_

Round 1
Reviewer 1 Report
I found this manuscript interesting and believe that immune activation is tightly regulated by the mesenchymal environment in close proximity to each other.
I have indeed some comments that mainly need to be addressed in the text.
1. Are the mesenchymal cells affected by the ongoing immune activation, I think this is a two-way approach. Therefore I wonder why not taking the extra EV from immune-induced content. Stimulated with LPS or immune factors like TNF. This would then be the greater likelihood that this EV would have a higher impact or?
2. Why no injections to Naive, this could have been injected in the same manner as the vehicle. Not several things that differ, but the CIA and no injections.
3. The vehicle is higher already from the start. Is this not a confounding factor? The treatment with EV starts first 6 days. Maybe just me who has a problem following the days. This should be discussed.
4. The day indications are confusing. The treatment starts Day 28 and so on from the immunization. But looking at the weight, arthritis development etc after treatment days, are they then different days after immunization? Then different days the mice were killed, How many each day and from which group and why? This needs to be clarified and using only one indicator of the days please explain.
4. In Figure 4 there are multiple comparisons but still just statistical just ttest This should be evaluated with an ANOVA.
5. The treatment is given systemic and therefore I would be interested in seeing some measurement of systemic inflammation. Weight of spleen or lymph nodes, CD86, CD163 in the spleen, or maybe the frequency of monocytes in either blood or spleen.
No direct problem with the English text. But please go through the text to find smaller errors and problems.
Reviewer 2 Report
I have read your article with interest. The authors have presented an excellent work which can potentially change the management of rhuematoid arthritis. I have a few minor suggestions to improve the readability of the manuscript.
1. Please use the full form of MSC-sEVs in the title.
2 Figure 5- please use arrows to point to inflammation, pannus, cartilage damage, etc., as it is not clear as presented currently.
3 Grammer:
Line 354: should have read 'Besides'
Line 355: After humoral factors, please insert a comma.
line 360: possess the ability- change to 'can'
Line 375: please remove the comma after 'modulation'
Minor English language editing is required as above.
